# Seven New Phenylhexanoids with Antioxidant Activity from *Saxifraga umbellulata* var. *pectinata*

**DOI:** 10.3390/molecules28093928

**Published:** 2023-05-06

**Authors:** Jiao Huang, Donglin Chen, Mengying Liu, Yarui Yu, Yi Zhang, Jing Huang

**Affiliations:** 1West China School of Pharmacy, Sichuan University, Chengdu 610041, China; 2School of Ethnic Medicine, Chengdu University of Traditional Chinese Medicine, Chengdu 611137, China

**Keywords:** *Saxifraga umbellulata* var. *pectinate*, phenylhexanoid, antioxidant activity

## Abstract

Seven new phenylhexanoids, (*S*)-(+)-3,4-dihydroxy-11-methoxyphenylhex-9-one (**1**), (*E*) 3,4-dihydroxy-phenylhex-10-en-9-one (**2**), (*E*)-4-hydroxyphenylhex-10-en-9-one (**3**), (*R*)-(−)-3,4,11-trihydroxyphenylhex-9-one 11-*O*-*β*-d-glucopyranoside (**4**), (*R*)-(−)-4,11-dihydroxyphenylhex-9-one 11-*O*-*β*-d-glucopyranoside (**5**), phenylhex-4,9,11-triol 11-*O*-*β*-d-glucopyranoside (**6**), and 9-*O*-acetyl-phenylhex-4,9,11-triol 11-*O*-*β*-d-glucopyranoside (**7**), were isolated and identified from Tibetan medicine *Saxifraga umbellulata* var. *pectinate*. The antioxidant activities of these compounds were evaluated using the DPPH and ABTS radical scavenging experiments. In the ABTS experiment, compounds **1** (IC_50_ 13.99 ± 2.53 μM) and **2** (IC_50_ 13.11 ± 0.94 μM) exhibited significantly better antioxidant activity than L-ascorbic acid (IC_50_ 23.51 ± 0.44 μM).

## 1. Introduction

*Saxifraga umbellulata* var. *pectinata*, which belongs to the family Saxifragaceae, is a perennial herb mainly distributed on plateaus above 3000 m above sea level [1]. It is one of the major varieties of the traditional Tibetan medicinal herb called ‘Songdi’ [2]. Tibetan clinical medicine analyzes ‘Songdi’ in terms of the four qi and the five tastes. It is believed that ‘Songdi’ is cold, and the taste is bitter, and it is mostly used for the treatment of hepatobiliary (Tripa disease) and digestive diseases (Tripa enteropathy) [3]. ‘Songdi’ has a long history of medicinal use in Tibetan medicine, and is mostly used as the ruling medicine or in combination with other Tibetan medicines in Tibetan medical prescriptions. For example, the classical Tibetan prescription for hepatitis and cholecystitis, Ershiwuwei Songshi pills, uses ‘Songdi’ as one of the prescribed drugs [3].

Model pharmaceutical research has demonstrated the antibacterial activity of ethanol extract, as well as the inhibitory effects on the proliferation of liver cancer (HepG2) cells of diarylnonanes, and the protective effect on peroxidative damage in L02 hepatocytes of flavonoids from *S. umbellulata* var. *pectinate* [4,5,6]. Phytochemistry studies have revealed the presence of diarylnonanes, flavonoids, triterpenoids, polyphenols, organic acids, and sterols in the plant of *S. umbellulata* var. *Pectinate* [6,7,8,9,10,11].

To investigate potential active ingredients for the treatment of hepatitis, cholecystitis, and digestive system diseases, additional studies of the phytochemistry and biology of *S. umbellulata* var. *pectinate* were conducted.

As a result, seven novel phenylhexanoids (Figure 1) were isolated and identified from *S. umbellulata* var. *pectinata*. This paper reports on the isolation, structural identification, and antioxidant effect of these compounds.

## 2. Results and Discussion

### 2.1. Identification of Compounds ***1**–**7***

**1** was obtained as a yellowish oil, and its molecular formula was analyzed as C_13_H_18_O_4_ by HR ESI MS (*m*/*z* 237.1132 [M-H]^−^, calculated 237.1127 for C_13_H_17_O_4_). This indicates five degrees of unsaturation.

The ^1^H NMR spectrum of **1** showed the following signals: a 1,2,4-trisubstituted benzene ring at δ_H_ 6.67 (1H, d, *J* = 8.0 Hz), 6.63 (1H, d, *J* = 2.1 Hz), and 6.50 (1H, dd, *J* = 8.0, 2.1 Hz), a methine group linking with oxygen at δ_H_ 3.78 (1H, dqd, *J* =7.5, 6.2, 5.2 Hz), three methylene groups at δ_H_ 2.72 (4H, m), 2.67 (1H, d, *J* = 15.9, 7.5 Hz), and 2.43 (1H, d, *J* = 15.9, 5.2 Hz), and a methyl group at δ_H_ 1.13 (3H, d, *J* = 6.2 Hz). In the ^13^C NMR spectrum of **1**, the following signals were shown: six benzene carbon signals at δ_C_ 116.3, 116.5, 120.6, 134.0, 144.5, and 146.2, six aliphatic carbon signals at δ_C_ 19.4, 30.0, 46.3, 50.6, 74.6, and 211.3, and a methoxy carbon signal at δ_C_ 56.8.

The above ^1^H and ^13^C NMR data (Table 1) of **1** indicate that there was a phenylhexyl skeleton in **1**, the same as in inonophenol A [12]. The molecular formula of **1** was C_13_H_18_O_4_ (*m*/*z* 238), having one more -CH_2_- group (*m*/*z* 14) than inonophenol A (C_12_H_16_O_4_, *m*/*z* 224). Comparing the ^1^H and ^13^C NMR data of **1** with the data for inonophenol A, most of the data are similar, except for the C_10_, C_11_, C_12_, and -OCH_3_ data of **1**.

In **1**, ^1^H and ^13^C NMR signals of an -OCH_3_ [δ_H_ 3.27 (3H, s); δ_C_ 56.8)] were observed. However, in inonophenol A, there was no ^1^H and ^13^C NMR signal of -OCH_3_.

The C_11_ signal (δ_C_ 74.6) in **1** was down-shifted by 9.5 chemical shift units compared to that in inonophenol A. Meanwhile, the C_10_ and C_12_ signals (δ_C_ 50.6 and 19.4) in **1** were up-shifted by 2.1 and 3.8 chemical shift units, respectively, compared to those in inonophenol A. These pieces of evidence prove that **1** resulted from the substitution of the hydroxyl group in inonophenol A with a methoxy group. The long-range correlations (Figure 2) of δ_H_ 3.27 (3H, s, -OCH_3_) with δ_C_ 74.6 (C_11_) and δ_H_ 3.78 (1H, dqd, *J* =7.5, 6.2, 5.2 Hz, 11-H) with δ_C_ 56.8 (-OCH_3_) in the HMBC spectrum of **1** further support this inference.

Based on the above MS, ^1^H NMR, ^13^C NMR, and HMBC spectra, it was confirmed that **1** is 3,4-dihydroxy-11-methoxyphenylhex-9-one.

The absolute configuration of C_11_ in **1** was determined to be in the *S*-configuration based on the similarity of the rotation value of **1** (${\left[\alpha \right]}_{D}^{25}$ + 6.29) to that of inonophenol A (${\left[\alpha \right]}_{D}^{20}$ + 5.42) [12]. Therefore, **1** was identified as (*S*)-(+)-3,4-dihydroxy-11-methoxy-phenylhex-9-one.

**2** was obtained as a yellow oil. Its molecular formula was determined to be C_12_H_14_O_3_ by HR ESI MS (*m*/*z* 205.0878 [M-H]^−^, calculated 205.0865 for C_12_H_13_O_3_), indicating six degrees of unsaturation.

The ^1^H NMR spectrum of **2** showed the following signals: a 1, 2, 4-trisubstituted benzene ring at δ_H_ 6.68 (1H, d, *J* = 8.0 Hz), 6.65 (1H, br.s), and 6.53 (1H, br.d, *J* = 8.0 Hz), a double bond with trans geometry at δ_H_ 6.94 (1H, dq, *J* = 15.7, 6.8 Hz) and 6.16 (1H, d, *J* = 15.7 Hz), two methylene groups at δ_H_ 2.85 (2H, t, *J* = 7.5 Hz) and 2.76 (2H, t, *J* = 7.5 Hz), and a methyl group at δ_H_ 1.91 (3H, d, *J* = 7.0 Hz). In the ^13^C NMR spectrum of **2**, the following signals were observed: six benzene carbon signals at δ_C_ 116.3, 116.5, 120.6, 134.1, 144.5, and 146.2, and six aliphatic carbon signals at δ_C_ 18.4, 30.8, 42.6, 132.8, 145.2, and 202.6.

The above ^1^H and ^13^C NMR data (Table 1) of **2** indicated that there was also a phenylhexyl skeleton in **2**, the same as in inonophenol A [12]. The molecular formula of **2** was C_12_H_14_O_3_ (*m*/*z* 206), which has one less -H_2_O group (*m*/*z* 18) than inonophenol A (C_12_H_16_O_4_, *m*/*z* 224). Comparing the ^1^H and ^13^C NMR data of **2** with the data for inonophenol A, most of the data are similar, except for the C_8_, C_9_, C_10_, C_11_, and C_12_ data of **2**.

In **2**, ^1^H and ^13^C NMR signals of a double bond with trans geometry [δ_H_ 6.94 (1H, dq, *J* = 15.7, 6.8 Hz) and 6.16 (1H, d, *J* = 15.7 Hz), δ_C_ 145.2 (C_11_) and 132.8 (C_10_)] were observed. However, in inonophenol A, there was no ^1^H and ^13^C NMR signal of a double bond. Instead, the NMR signals of a methylene group [δ_H_ 2.48 (1H, dd, *J* = 16.0, 4.9 Hz) and 2.56 (1H, dd, *J* = 1.60, 7.8 Hz), δ_C_ 52.7 (C_10_)] and a methine group linking with oxygen [δ_H_ 4.17 (1H, m), δ_C_ 65.1 (C_11_)] were observed in inonophenol A. In addition, meanwhile, the C_8_ (δ_C_ 42.6), C_9_ (δ_C_ 202.6), and C_12_ (δ_C_ 18.4) signals in **2** were up-shifted by 3.7, 9.3, and 5.1 chemical shift units, respectively, compared to those in inonophenol A. 

The HMBC spectrum of **2** showed the long-range correlations (Figure 2) of δ_H_ 6.16 (1H, d, *J* = 15.7 Hz, 10-H) with δ_C_ 18.4 (C_12_) and 42.6 (C_8_), and δ_H_ 6.94 (1H, d, *J* = 15.7 Hz, 6.8 Hz, 11-H) with δ_C_ 202.6 (C_9_). The above MS, ^1^H, ^13^C NMR, and HMBC spectra, **2** was a dehydration product of inonophenol A and identified as (*E*)3,4-dihydroxy-phenylhex-10-en-9-one.

**3** was obtained as a yellow oil, and its molecular formula was analyzed as C_12_H_14_O_3_ by HR ESI MS (*m*/*z* 189.0933 [M-H]^−^, calculated 189.0916 for C_12_H_13_O_3_), indicating six degrees of unsaturation.

The ^1^H NMR spectrum of **3** showed the following signals: a *p*-substituted benzene ring at δ_H_ 6.68 and 7.00 (2H each, d, *J* = 8.5 Hz); a double bond with trans geometry at δ_H_ 6.90 (1H, dq, *J* = 15.8, 6.8 Hz) and 6.11 (1H, d, *J* = 15.8, 1.6 Hz), two methylene groups at δ_H_ 2.80 (4H, m), and a methyl group at δ_H_ 1.87 (3H, dd, *J* = 6.8, 1.6Hz). In the ^13^C NMR spectrum of **3**, the following signals were shown: six benzene ring carbon signals at δ_C_ 116.1, 116.1, 130.3, 130.3, 133.2, and 156.6, and six aliphatic carbon signals at δ_C_ 18.4, 30.4, 42.6, 132.8, 145.2 and 202.5.

The above ^1^H and ^13^C NMR data (Table 1) of **3** indicated that there was also a phenylhexyl skeleton in **3**, the same as in **2**. The molecular formula of **3** was C_12_H_14_O_3_ (*m*/*z* 190), which was one less -*O*- atom (*m*/*z* 16) than **2** (C_12_H_14_O_3_, *m*/*z* 206). Comparing the ^1^H and ^13^C NMR data of **3** with the data for **2**, most of the data are similar, except for the benzene data of **3**. The ^1^H and ^13^C NMR signals of a *p*-substituted phenyl [δ_H_ 6.68, 7.00 (2H each, d, *J* = 8.5 Hz); δ_C_ 116.1–156.6 (C_1_–C_6_)] were observed in **3**. However, in **2**, there was no ^1^H and ^13^C NMR signal of *p*-substituted phenyl, instead, the NMR data of a 1, 2, 4-trisubstituted phenyl [δ_H_ 6.68 (1H, d, *J* = 8.0 Hz), 6.65 (1H, br.s), and 6.53 (1H, br.d, *J* = 8.0 Hz); δ_C_ 116.3–146.2 (C_1_–C_6_)] were observed. This evidence proved that **3** was a dehydroxyl product of **2**. The long-range correlations (Figure 2) of δ_H_ 7.00 (2H, d, *J* = 8.5 Hz, 2-H/6-H) with δ_C_ 30.4 (C_7_) and 156.6 (C_4_), and δ_H_ 6.68 (2H, d, *J* = 8.5 Hz, 3-H/5-H) with δ_C_ 133.2 (C_1_) in the HMBC spectrum of **3** also proved the above inference.

Based on the above MS, ^1^H, ^13^C NMR, and HMBC spectra, it was confirmed that **3** was (*E*)-4-hydroxy-phenylhex-10-en-9-one.

**4** was obtained as a white amorphous powder, and its molecular formula was analyzed as C_18_H_26_O_9_ by HR ESI MS (*m*/*z* 385.1515 [M-H]^−^, calculated 385.1499 for C_18_H_25_O_9_), indicating six degrees of unsaturation.

The ^1^H NMR spectrum of **4** showed the following signals: a 1, 2, 4-trisubstituted benzene ring at δ_H_ 6.67 (1H, d, *J* = 8.0 Hz), 6.64 (1H, d, *J* = 2.1 Hz), and 6.52 (1H, dd, *J* = 8.0, 2.1 Hz); a methine group linking with oxygen at δ_H_ 4.34 (1H, m); three methylene groups at δ_H_ 2.80 (2H, m), 2.72 (2H, m), 2.83 (1H, m) and 2.53 (1H, dd, *J* = 15.9, 5.4 Hz); and a methyl group at δ_H_ 1.20 (3H, d, *J* = 6.2 Hz). In the ^13^C NMR spectrum of **4**, the following signals were shown: six benzene ring carbon signals at δ_C_ 116.3, 116.5, 120.5, 134.1, 144.4, and 146.1, and six aliphatic carbon signals at δ_C_ 20.3, 30.0, 46.2, 51.6, 72.4, and 211.9.

The above ^1^H and ^13^C NMR data (Table 2) of **4** indicate that there was a phenylhexyl skeleton in **4**, the same as in inonophenol A [12]. The molecular formula of **4** was C_18_H_26_O_9_ (*m*/*z* 386), which was one more -C_6_H_10_O_5_- group (*m*/*z* 162) than inonophenol A (C_12_H_16_O_4_, *m*/*z* 224). Comparing the ^1^H and ^13^C NMR data of **4** with data for inonophenol A, most of the data are similar, except for the C_10_, C_11_, C_12_, and -C_6_H_10_O_5_- data of **4**.

In **4**, the ^1^H and ^13^C NMR signals of a monosaccharide moiety [δ_H_ 4.34 (1H, d, *J* = 7.2 Hz) and 3.12–3.84; δ_C_ 102.4 and 62.9–78.0 (C_1′_-C_6′_)] were observed. However, in inonophenol A, there were no ^1^H and ^13^C NMR signals of this.

The acid hydrolysis experiment on **4** afforded D-glucose, confirmed by TLC and a comparison of its NMR data with those of (5*S*)-1,7-bis-(3,4-dihydroxy-phenyl)-5-hydroxyheptan-3-one-5-*O-β*-d-glucopyranoside [13], and the relative configuration of the anomeric carbon to be *β*-configuration due to its large coupling constant. Based on the above evidence, the monosaccharide was determined to be *β*-d-glucopyranose.

The C_11_ signal (δ_C_ 72.4) in **4** was down-shifted by 7.3 chemical shift units compared to that in inonophenol A. Meanwhile, the C_10_ and C_12_ signals (δ_C_ 51.6 and 20.3) in **4** were up-shifted by 1.1 and 3.2 chemical shift units, respectively, compared to those in inonophenol A. This evidence proved that **4** was a glycation product of inonophenol A by a *β*-d-glucopyranose moiety. The long-range correlations (Figure 2) of δ_H_ 4.34 (1H, d, *J* = 7.2 Hz, 1′-H) with δ_C_ 72.4 (C_11_), and δ_H_ 4.34 (1H, m, 11-H) with δ_C_ 102.4 (C_1′_) in the HMBC spectrum of **4** further support this inference. Based on the above MS, ^1^H, ^13^C NMR, and HMBC spectra, it was confirmed that **4** is 3,4,11-trihydroxyphenylhex-9-one 11-*O*-*β*-d-glucopyranoside.

The absolute configuration of C_11_ in **4** was determined to be *R*-configuration based on the contrast of the rotation value of the hydrolyzed aglycone (${\left[\alpha \right]}_{D}^{25}$ − 4.56) of **4** with that of inonophenol A (${\left[\alpha \right]}_{D}^{20}$ + 5.42) [12]. Therefore, **4** was identified as (*R*)-(−)-3,4,11-trihydroxyphenylhex-9-one 11-*O*-*β*-d-glucopyranoside.

**5** was obtained as a white amorphous powder, and its molecular formula was analyzed as C_18_H_26_O_8_ by HR ESI MS (*m*/*z* 369.1527 [M-H]^−^, calculated 369.1549 for C_18_H_25_O_8_), indicating six degrees of unsaturation.

The ^1^H NMR spectrum of **5** showed the following signals: a *p*-substituted benzene ring at δ_H_ 6.70 and 7.02 (2H each, d, *J* = 8.5 Hz); a methine group linking with oxygen at δ_H_ 4.34 (1H, m); three methylene groups at δ_H_ 2.80 (2H, m), 2.77 (2H, m), 2.84 (1H, dd, *J =* 15.9, 7.3 Hz) and 2.53 (1H, dd, *J =* 15.9, 5.4 Hz); a methyl group at δ_H_ 1.20 (3H, d, *J* = 6.2 Hz); an anomeric proton at δ_H_ 4.34 (1H, d, *J* = 7.7 Hz); and a typical sugar moiety proton at δ_H_ 3.12–3.84. In the ^13^C NMR spectrum of **5**, the following signals were shown: six benzene ring carbon signals at δ_C_ 116.1, 116.1, 130.3, 130.3, 133.3, and 156.5, six aliphatic carbon signals at δ_C_ 20.3, 29.7, 49.2, 51.5, 72.3, and 211.9, and typical sugar moiety carbon signals at δ_C_ 62.8, 71.7, 75.0, 77.8, 78.0 and 102.3.

The above ^1^H and ^13^C NMR data (Table 2) of **5** indicated that there was also a phenylhexyl glycoside skeleton in **5**, the same as in **4**. The molecular formula of **5** was C_18_H_26_O_8_ (*m*/*z* 370), which was one less -*O*- atom (*m*/*z* 16) than **4** (C_18_H_26_O_9_, *m*/*z* 386). Comparing the ^1^H and ^13^C NMR data of **5** with the data for **4**, most of the data are similar, except for the benzene data of **5**. The ^1^H and ^13^C NMR signals of a *p*-substituted phenyl [δ_H_ 6.70, 7.02 (2H each, d, *J* = 8.0 Hz); δ_C_ 116.1–156.5 (C_1_–C_6_)] were observed in **5**. However, in **4**, there was no ^1^H and ^13^C NMR signal of *p*-substituted phenyl, instead, the NMR data of a 1, 2, 4-trisubstituted phenyl [δ_H_ 6.67 (1H, d, *J* = 8.0 Hz), 6.64 (1H, d, *J* = 2.1 Hz), and 6.52 (1H, dd, *J* = 8.0, 2.1 Hz); δ_C_ 116.3–146.1 (C_1_–C_6_)] were observed. This evidence proved that **5** was a dehydroxyl product of **4**.

The long-range correlations (Figure 2) of δ_H_ 7.02 (2H, d, *J* = 8.5 Hz, 2-H/6-H) with δ_C_ 29.7 (C_7_) and 156.3 (C_4_), and δ_H_ 6.70 (2H, d, *J* = 8.5 Hz, 3-H/5-H) with δ_C_ 133.2 (C_1_) in the HMBC spectrum of **5** also proved the above inference.

Based on the above MS, ^1^H, ^13^C NMR, and HMBC spectra, **5** was identified as being 4,11-dihydroxyphenylhex-9-one 11-*O*-*β*-d-glucopyranoside.

The absolute configuration of C_11_ in **5** was determined to be *R*-configuration based on the similarity of the rotation value of **5** (${\left[\alpha \right]}_{D}^{25}$ − 19.6) to that of **4** (${\left[\alpha \right]}_{D}^{20}$ − 22.3). Therefore, **5** was identified as (*R*)-(−)-4,11-dihydroxy-phenylhex-9-one 11-*O*-*β*-d-glucopyranoside.

**6** was afforded as a white amorphous powder, and its molecular formula was analyzed as C_18_H_28_O_8_ by HR ESI MS (*m*/*z* 371.1705 [M-H]^−^, calculated 371.1706 for C_18_H_27_O_8_), indicating five degrees of unsaturation.

The ^1^H NMR spectrum of **6** showed the following signals: a *p*-substituted benzene ring at δ_H_ 6.70 and 7.03 (2H each, d, *J* = 8.5 Hz); two methine groups linking with oxygen at δ_H_ 3.74 (1H, tt, *J =* 8.5, 4.3 Hz) and 4.10 (1H, m); three methylene groups at δ_H_ 2.62 (2H, m), 1.71 (2H, m), 1.84 (1H, m) and 1.58 (1H, m); a methyl group at δ_H_ 1.21 (3H, d, *J* = 6.1 Hz); an anomeric proton at δ_H_ 4.36 (1H, d, *J* = 7.8 Hz); and a typical sugar moiety proton at δ_H_ 3.15–3.87. In the ^13^C NMR spectrum of **6**, the following signals were shown: six benzene ring carbon signals at δ_C_ 116.1, 116.1, 130.3, 130.3, 133.3, and 156.5, six aliphatic carbon signals at δ_C_ 20.1, 32.0, 40.8, 45.5, 70.1, and 74.3, and typical sugar moiety carbon signals at δ_C_ 62.9, 71.7, 75.1, 77.9, 78.0 and 102.3.

The above ^1^H and ^13^C NMR data (Table 2) of **6** indicated that there was also a phenylhexyl glycoside skeleton in **6**, the same as in **5**. The molecular formula of **6** was C_18_H_28_O_8_ (*m*/*z* 372), which was two -H- atoms (*m*/*z* 2) more than **5** (C_18_H_26_O_8_, *m*/*z* 370). Comparing the ^1^H and ^13^C NMR data for **6** with data for **5**, most of the data are similar, except for the C_8_, C_9_, and C_10_ data of **6**.

In **5**, the ^13^C NMR signal [δ_C_ 211.9] of a C=O was observed. However, in **6**, there was no ^13^C NMR signal of a C=O group. Instead, the ^1^H and ^13^C NMR signals [δ_H_ 3.74 (1H, tt, *J =* 8.5, 4.3 Hz); δ_C_ 70.1 (C_9_)] of one more methine group linking with oxygen were observed; meanwhile, the ^13^C signal of C_9_ (δ_C_ 70.1) in **6** was up-shifted by 141.8 and the ^13^C signals of C_8_ and C_10_ (δ_C_ 40.8 and 45.5) were down-shifted by 5.4 and 6.0 chemical shift units, respectively, compared to those in **5**. This evidence proves that **6** should be the product of the reduction of the carbonyl group in **5**. The long-range correlations (Figure 2) of δ_H_ 2.62 (2H, m, 7-H) and 4.10 (H, m, 11-H) with δ_C_ 70.1 (C_9_) in the HMBC spectrum of **6** further support this inference.

Based on the above MS, ^1^H, ^13^C NMR, and HMBC spectra, it was confirmed that **6** was phenylhex-4,9,11-triol 11-*O*-*β*-d-glucopyranoside. Due to technical limitations, the absolute configuration of **6** could not be determined.

**7** was afforded as a white amorphous powder, and its molecular formula was analyzed as C_20_H_30_O_9_ by HR-ESI-MS (*m*/*z* 413.1804 [M-H]^−^, calculated 413.1812 for C_20_H_29_O_9_), indicating six degrees of unsaturation.

The ^1^H NMR spectrum of **1** showed the following signals: a *p*-substituted benzene ring at δ_H_ 7.01 and 6.69 (2H each, d, *J* = 8.0 Hz); two methine groups linking with oxygen at δ_H_ 5.08 (1H, m) and 3.97 (1H, dq, *J* = 7.8, 6.0 Hz); three methylene groups at δ_H_ 2.55 (2H, m), 1.97, 1.68 (1H each, m), 1.95, 1.84 (1H each, m); two methyl groups at δ_H_ 2.01 (3H, s) and 1.19 (3H, d, *J* = 6.0 Hz); an anomeric proton at δ_H_ 4.32 (1H, d, *J* = 7.7 Hz); and a typical sugar moiety proton at δ_H_ 3.14–3.79. In the ^13^C NMR spectrum of **7**, the following signals were shown: six benzene ring carbon signals at δ_C_ 116.1, 116.1, 130.3, 130.3, 133.8, and 156.4, six aliphatic carbon signals at δ_C_ 20.0, 31.6, 37.0, 42.5, 72.7 and 73.3, and typical sugar moiety carbon signals at δ_C_ 62.9, 71.7, 75.1, 77.8, 78.0 and 101.8.

The ^1^H and ^13^C NMR data (Table 2) of **7** indicated that there was a phenylhexyl glycoside skeleton in **7**, the same as in **6**. The molecular formula of **7** was C_20_H_30_O_9_ (*m*/*z* 414), which was one more -COCH_2_- group (*m*/*z* 42) than **6** (C_18_H_27_O_8_, *m*/*z* 372). Comparing the ^1^H and ^13^C NMR data of **7** with data for **6**, most of the data are similar, except for the C_8_, C_9_, C_10_, and -COCH_3_ data of **7**.

In **7**, the ^1^H and ^13^C NMR signals of a -COCH_3_ [δ_H_ 2.01 (3H, s); δ_C_ 173.0 and 21.3] were observed. However, in **6**, there was no ^1^H and ^13^C NMR signal of -COCH_3_.

The C_8_, C_9_, and C_10_ signals (δ_C_ 37.0, 72.7, and 42.5) in **7** were up-shifted 1.6, 3.8, and 3.0 chemical shift units, respectively, compared to those in **6**. These pieces of evidence prove that **7** should be the substitution product of the C_9_-OH in **6** by C_9_-OCOCH_3_. The long-range correlations (Figure 2) of δ_H_ 2.01 (3H, s, 2″-H) with δ_C_ 173.0 (C_1″_) and δ_H_ 5.08 (1H, m, 9-H) with δ_C_ 31.6 (C_7_), 72.7 (C_11_), and 173.0 (C_1″_) in the HMBC spectrum of **7** further support this inference.

Based on the above MS, ^1^H, ^13^C NMR, and HMBC spectra, it was confirmed that **7** is 9-*O*-acetylphenylhex-4,9,11-triol 11-*O*-*β*-d-glucopyranoside. Due to technical limitations, the absolute configuration of **7** could not be determined.

### 2.2. The Antioxidant Activities of Compounds ***1**–**7***

Compounds **1**–**7** isolated from the title plant were tested for their antioxidant effects. The results of the antioxidant activity assays are listed in Table 3.

The 2,2′-azino-bis (3-ethylbenzthiazoline-6-sulphonic acid) ammonium salt (ABTS) radical scavenging effects of compounds **1** (IC_50_ 13.99 ± 2.53 μM) and **2** (IC_50_ 13.11 ± 0.94 μM) were more potent than the positive control, L-(+)-ascorbic acid (IC_50_ 23.51 ± 0.44 μM) (*p* < 0.05), while the ABTS radical scavenging effects of compounds **4** (IC_50_ 28.44 ± 3.86 μM) and **7** (IC_50_ 27.03 ± 0.55 μM) were equivalent to L-(+)-ascorbic acid (*p* > 0.05).

The results of the DPPH and ABTS assays showed that the catechol groups of compounds are very important for enhancing activity. During the normal metabolic process of living organisms, more chemically active oxygen-containing substances, also known as reactive oxygen species (ROS), are produced [14]. Low levels of ROS are essential for a variety of biological functions, such as cell survival, growth, proliferation, differentiation and immune response [15]. When the generation of reactive oxygen radicals is higher than the antioxidant capacity, oxidative stress (OS) occurs. Excess ROS can cause damage to proteins, DNA and RNA, leading to genetic alterations in cells and promoting the development of disease or cell death [16]. Numerous studies have shown that cardiovascular diseases, inflammation, malignant tumors, diabetes, and atherosclerosis are all related to oxidative damage in the body, which is caused by excess free radicals or ROS generated during metabolic processes [17]. Therefore, drugs with the ability to scavenge reactive oxygen radicals have an important role in the pathogenesis of inflammation-related diseases due to oxidative stress caused by excess oxygen free radicals. **1**–**7** might be among the active constituents of *S. umbellulata* var. *pectinata* that play a role in the treatment of liver inflammation-associated diseases.

## 3. Materials and Methods

### 3.1. General Experimental Procedure

NMR spectra were obtained using an AC-E200 400 NMR spectrometer (Bruker Corporation, German) (^1^H at 400 MHz, ^13^C at 100 MHz) with CD_3_OD as the solvent at 25 °C, using TMS as the internal standard. The UV spectrum was obtained using a UV3600 spectrophotometer (Shanghai Pharmaceutical Machinery Co., Ltd., Shanghai, China). The IR absorption spectrum was recorded with a Nicolet 6700 spectrophotometer (Thermo Electron Co., Waltham, MA, USA). High-resolution electrospray ionization mass spectroscopy (HR ESI MS) was performed on a Waters Xevo G2-XS Q-TOF Premier mass spectrometer (Waters, Milford, MA, USA). The optical rotation value was tested at room temperature with a JASCO P-1020 polarimeter (Jasco Co., Tokyo, Japan). The microplate reader used in the antioxidant activity experiment was a SparkTM 10 M (Tecan Co., Männedorf, Switzerland).

Column chromatography (CC) was performed using silica gel (100–200 and 300–400 mesh; Qingdao Marine Chemical Factory, Qingdao, China), polyamide (60–90 mesh, Jiangsu Changfeng Chemical Industry Co., Yangzhou, China), RP-C18 silica gel (20–45 μm; Mitsubishi Chemical Co., Tokyo, Japan), and Sephadex LH-20 (40–70 μm; Amersham Pharmacia Biotech, Stockholm, Sweden). TLC was carried out using HPTLC Fertigplatten Kieselgel 60 F_254_ plates (Merck, Darmstadt, Germany), which were sprayed with the *α*-naphthol–sulfuric acid solution or 10% sulfuric acid–ethanolic solution and then baked for 3–5 min at a temperature of 105 °C. UV-vis absorbance was measured with a UV2700 spectrophotometer (Shimadzu, Kyoto, Japan). 2,2-Diphenyl-1-picrylhydrazyl (DPPH) was acquired from Macklin Biochemical Co., Ltd. (Shanghai, China). 2,2′-azinobis (3-ethylbenz thiazoline-6-sulphonic acid) ammonium salt (ABTS) was obtained from Aladdin Industrial Co., Ltd. (Shanghai, China).

### 3.2. Plant Material

The whole plant of *S. umbellulata* var. *pectinate* was collected from Tibet, China, in July 2020, and confirmed by Prof. Yi Zhang (School of Ethnic Medicine, Chengdu University of Traditional Chinese Medicine, Chengdu 611137, China). The specimen (No. BCHEC 20200912) was deposited in the School of Ethnic Medicine, Chengdu University of Traditional Chinese Medicine, Chengdu 611137, China.

### 3.3. Extraction and Isolation

The dried crude powder of the title plant (10 kg) was extracted with 95% ethanol (100 L) at room temperature three times (every 7 days). The ethanol extract was filtered and condensed in vacuo to yield ethanol extract (1.6 kg). The ethanol extract (1.6 kg) was mixed with silica gel (100–200 mesh) at a ratio of 1:1, then put into a continuous extractor and extracted with petroleum ether, dichloromethane, ethyl acetate, and methanol, respectively, and petroleum ether extract (245.6 g), dichloromethane extract (169.5 g), ethyl acetate extract (183.0 g), and methanol extract (1011.0 g) were obtained by depressurization and concentration.

The methanol extract (400 g) was separated on a silica gel column (CH_2_Cl_2_-MeOH 20:1–0:1) to obtain 8 fractions (Frs.1–8), according to the TLC analysis. Fr.2 (7.6 g) was separated on a silica gel column (CH_2_Cl_2_-MeOH, 120:1–65:1) to obtain 6 fractions (Frs.2–1 to 2–6). Fr.2–1 (218 mg) was purified using an RP-18 reverse-phase chromatography column (MeOH-H_2_O, 0:1–2:1) and a Sephadex LH-20 gel chromatography column (CH_2_Cl_2_-MeOH 1:1) to yield compound **3** (24 mg). Fr.2–4 (305 mg) was purified using an RP-18 reverse-phase chromatography column (MeOH-H_2_O, 1:4–3:1) and a Sephadex LH-20 gel chromatography column (CH_2_Cl_2_-MeOH 1:1) to yield compounds **1** (10 mg) and **2** (6 mg). Fr.4 (30 g) was separated using a polyamide chromatography column (EtOH-H_2_O, 0:10–4:1) to obtain 5 fractions (Frs.4–1 to 4–5). Fr.4–1 (8.0 g) was separated using a silica gel chromatography column (CH_2_Cl_2_-MeOH, 60:1–1:1) to obtain 5 fractions (Frs. 4–1-1 to 1–5). Fr.4–1-2 (110 mg) was purified using an RP-18 reverse-phase chromatography column (MeOH-H_2_O, 0:1–1:1) and a Sephadex LH-20 gel chromatography column (CH_2_Cl_2_-MeOH 1:1) to obtain compounds **5** (34 mg) and **7** (6 mg). Fr.4–1-4 (700 mg) was purified using an RP-18 reverse-phase chromatography column (MeOH-H_2_O, 0:1–1:2) and Sephadex LH-20 gel chromatography column (CH_2_Cl_2_-MeOH 1:1) to obtain compounds **4** (18 mg) and **6** (18 mg).

Compound **1**: yellowish oil. ${\left[\alpha \right]}_{D}^{25}$ + 6.29 (c 0.03, MeOH). UV (MeOH) λ_max_ (log ε): 284 (3.57) nm, IR (KBr) υ_max_: 3380, 2937, 1706, 1605, 1519, 1445 cm^−1^; ^1^H NMR (CD_3_OD, 400 MHz) and ^13^C NMR (CD_3_OD, 100 MHz) data, see Table 1; HR ESI MS *m*/*z* 237.1132 [M-H]^−^ (calculated for C_13_H_17_O_4_, 237.1127).

Compound **2**: yellowish oil. UV (MeOH) λ_max_ (log ε): 282 (3.50) nm, IR (KBr) υ_max_: 3381, 2927, 1658, 1628, 1520, 1443 cm^−1^; ^1^H NMR (CD_3_OD, 400 MHz) and ^13^C NMR (CD_3_OD, 100 MHz) data, see Table 1; HR ESI MS *m*/*z* 205.0878 [M-H]^−^ (calculated for C_12_H_13_O_3_, 205.0865).

Compound **3**: yellowish oil. UV (MeOH) λ_max_ (log ε): 282 (3.55) nm, IR (KBr) υ_max_: 3332, 1658, 1615, 1516, 1442 cm^−1^; ^1^H NMR (CD_3_OD, 400 MHz) and ^13^C NMR (CD_3_OD, 100 MHz) data, see Table 1; HR ESI MS *m*/*z* 189.0933 [M-H]^−^ (calculated for C_12_H_13_O_2_, 189.0916).

Compound **4**: white amorphous powder. ${\left[\alpha \right]}_{D}^{25}$ −22.3 (c 0.06, MeOH). UV(MeOH) λ_max_ (log ε): 284 (3.42) nm, IR (KBr) υ_max_: 3369, 2925, 1702, 1520, 1447 cm^−1^; ^1^H NMR (CD_3_OD, 400 MHz) and ^13^C NMR (CD_3_OD, 100 MHz) data, see Table 2; HR ESI MS *m*/*z* 385.1515 [M-H]^−^ (calculated for C_18_H_25_O_9_, 385.1499).

Compound **5**: white amorphous powder. ${\left[\alpha \right]}_{D}^{25}$ −19.6 (c 0.11, MeOH). UV (MeOH) λ_max_ (log ε): 280 (3.37) nm, IR (KBr) υ_max_: 3365, 2916, 1704, 1614, 1516, 1449 cm^−1^; ^1^H NMR (CD_3_OD, 400 MHz) and ^13^C NMR (CD_3_OD, 100 MHz) data, see Table 2; HR ESI MS *m*/*z* 369.1527 [M-H]^−^ (calculated for C_18_H_25_O_8_, 369.1549).

Compound **6**: white amorphous powder. ${\left[\alpha \right]}_{D}^{25}$ −35.0 (c 0.02, MeOH). UV (MeOH) λ_max_ (log ε): 280 (3.35) nm, IR (KBr) υ_max_: 3381, 2927, 1613, 1516, 1452 cm^−1^; ^1^H NMR (CD_3_OD, 400 MHz) and ^13^C NMR (CD_3_OD, 100 MHz) data, see Table 2; HR ESI MS *m*/*z* 371.1705 [M-H]^−^ (calculated for C_18_H_27_O_8_, 371.1706).

Compound **7**: white amorphous powder. ${\left[\alpha \right]}_{D}^{25}$ −24.0 (c 0.06, MeOH). UV (MeOH) λ_max_ (log ε): 280 (3.45), IR (KBr) υ_max_: 3363, 2920, 1711, 1516, 1451 cm^−1^; ^1^H NMR (CD_3_OD, 400 MHz) and ^13^C NMR (CD_3_OD, 100 MHz) data, see Table 2; HR ESI MS *m*/*z* 413.1804 [M-H]^−^ (calculated for C_20_H_29_O_9_, 413.1812).

### 3.4. Acid Hydrolysis of Compound ***4***

Dissolved **4** (4 mg) with 0.1 mL CH_3_OH was added to 4 mL of H_2_SO_4_ aqueous solution (1 mol/L) and kept at 90 °C for 3 h. Adjusted the reaction solution to pH neutral with sodium hydroxide solution (1 mol/L), and then ethyl acetate eluate was added to extract the solution 3 times. An ethyl acetate phase and an aqueous phase were obtained. The aqueous phase permeated and condensed, and monosaccharides in the concentrated solution were confirmed by TLC (CHCl_3_-CH_3_OH-H_2_O = 3:2:0.1) and D-glucose (standard sample) [18]. The *Rf* value of D-glucose was 0.6.

### 3.5. Determination of Antioxidant Activity

DPPH and ABTS radical scavenging experiments were performed to measure the antioxidant activity of compounds **1**–**7** [19,20].

#### 3.5.1. DPPH Radical Scavenging Assay

A 100 μL volume of DPPH anhydrous ethanol solution (120 μM) was added to 100 μL anhydrous ethanol sample solution (12.5, 25, 50, 100, 200, and 400 μM) in a 96-well plate. The mixture was allowed to react at room temperature for 30 min in the dark, and then the absorbance of the mixture at a wavelength of 517 nm was measured with a microplate reader. Three parallel experiments were conducted. DPPH radical scavenging activity was calculated using the following formula: DPPH scavenging activity was calculated by the following formula: DPPH scavenging activity (%) = (*A*_control_ − *A*_sample_)/*A*_control_ × 100%, where *A*_control_ is the absorbance of the anhydrous ethanol control without samples, and *A*_sample_ is the absorbance of sample. L-ascorbic acid was used as a positive control in the experiment.

#### 3.5.2. ABTS Radical Scavenging Assay

A 100 μL volume of ABTS anhydrous ethanol solution (140 μM) was added to 100 μL anhydrous ethanol sample solution (12.5, 25, 50, 100, 200, and 400 μM) in a 96-well plate. The mixture was reacted at room temperature for 5 min in the dark, and then the absorbance of the mixture at a wavelength of 734 nm was measured with a microplate reader. Three parallel experiments were performed. The ABTS radical scavenging activity was calculated by the following formula: ABTS scavenging activity (%) = (*A*_control_ − *A*_sample_)/*A*_control_ × 100%, where *A*_control_ is the absorbance of anhydrous ethanol control without samples, and *A*_sample_ is the absorbance of the sample. L-ascorbic acid was used as a positive control in the experiment.

### 3.6. Statistical Analyses

The statistical analyses were performed using GraphPad Prism 8.0. Every sample was analyzed in triplicate. The IC_50_ value of a compound (where half of DPPH and ABTS free radicals are cleared) was obtained by plotting the scavenging percentage of every sample of the compound against its concentration. The results are expressed as the mean ± standard deviation (SD). The difference in the means between compound and positive control was analyzed by one-way analysis of variance (ANOVA) using SPSS 25.0, to judge whether there was a statistically significant difference between the groups (*p* < 0.05).

## 4. Conclusions

To date, only three similar components (inositol A, inositol B, and hispolon) similar to the skeleton of the compounds reported in this paper have been reported to have been isolated from microorganisms [12,21]. However, the original literature defines them as being of the phenyl-substituted hexane type.

This paper reports these compounds from plants for the first time, and based on the skeleton naming rule of natural products, the skeleton type was denoted as phenylhexanoid. This name is more in line with its biosynthetic pathway.

Although compounds **1**–**7** showed some antioxidant activities, further research is needed to see if they have an effect when used for the treatment of hepatitis, cholecystitis, and digestive diseases.

## Figures and Tables

**Figure 1 molecules-28-03928-f001:**
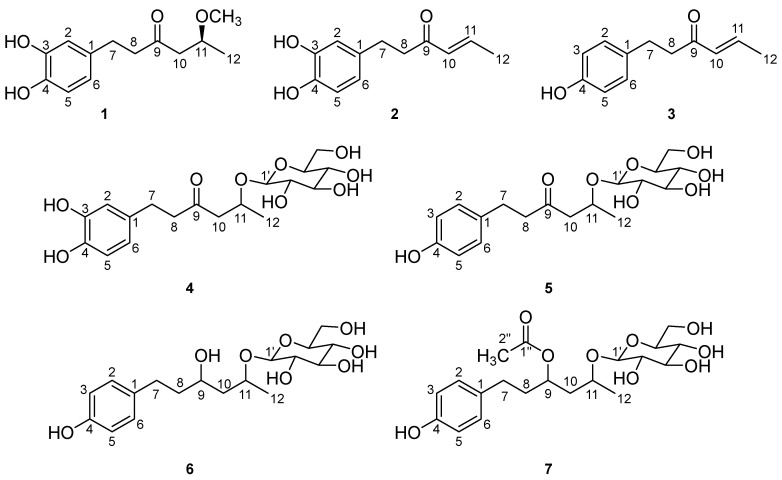
Structures of compounds **1**–**7**.

**Figure 2 molecules-28-03928-f002:**
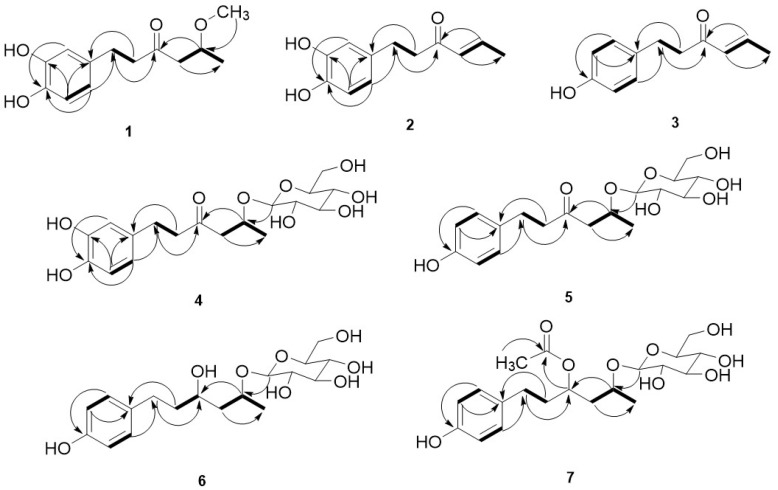
The key ^1^H-^1^H COSY (bold) and HMBC (H→C) correlations of **1**–**7**.

**Table 1 molecules-28-03928-t001:** ^1^H NMR (400 MHz) and ^13^C NMR (100 MHz) data of **1**–**3** in CD_3_OD.

Position	1		2		3	
	δ_C_	δ_H_ (*J* in Hz)	δ_C_	δ_H_ (*J* in Hz)	δ_C_	δ_H_ (*J* in Hz)
1	134.0		134.1		133.2	
2	116.3	6.63 d (2.1)	116.3	6.65 br.s	130.3	7.00 d (8.5)
3	146.2		146.2		116.1	6.68 d (8.5)
4	144.5		144.5		156.6	
5	116.5	6.67 d (8.0)	116.5	6.68 d (8.0)	116.1	6.68 d (8.5)
6	120.6	6.50 dd (8.0, 2.1)	120.6	6.53 br.d (8.0)	130.3	7.00 d (8.5)
7	30.0	2.72 t (4.0)	30.8	2.76 t (7.5)	30.4	2.79 m
8	46.3	2.72 t (4.0)	42.6	2.85 t (7.5)	42.6	2.79 m
9	211.3		202.6		202.5	
10	50.6	2.67 dd (15.9, 7.5) 2.43 dd (15.9, 5.2)	132.8	6.16 d (15.7)	132.8	6.11 dq (15.8, 1.6)
11	74.6	3.78 dqd (7.5, 6.2, 5.2)	145.2	6.94 dq (14.1, 6.8)	145.2	6.89 dq (15.8, 6.8)
12	19.4	1.13 d (6.2)	18.4	1.19 d (7.0)	18.4	1.86 dd (6.8, 1.7)
-OCO_3_	56.8	3.27 s				

**Table 2 molecules-28-03928-t002:** ^1^H NMR (400 MHz) and ^13^C NMR (100 MHz) data of **4**–**7** in CD_3_OD.

Position	4		5		6		7	
	δ_C_	δ_H_ (*J* in Hz)	δ_C_	δ_H_ (*J* in Hz)	δ_C_	δ_H_ (*J* in Hz)	δ_C_	δ_H_ (*J* in Hz)
1	134.1		133.3		134.5		133.8	
2	116.3	6.64 d (2.0)	130.3	7.02 d (8.0)	130.3	7.03 d (8.5)	130.3	7.01 d (8.0)
3	146.1		116.1	6.70 d (8.0)	116.7	6.70 d (8.5)	116.1	6.69 d (8.0)
4	144.4		156.5		156.3		156.4	
5	116.5	6.67 d (8.0)	116.1	6.70 d (8.0)	116.7	6.70 d (8.5)	116.1	6.69 d (8.0)
6	120.5	6.52 dd (8.0, 2.0)	130.3	7.02 d (8.0)	130.3	7.03 d (8.5)	130.3	7.01 d (8.0)
7	30.0	2.72 m	29.7	2.77 m	32.0	2.62 m	31.6	2.55 m
8	46.2	2.80 m	46.2	2.80 m	40.8	1.71 m	37.0	1.95 m 1.84 m
9	211.9		211.9		70.1	3.74 tt (8.5, 4.3)	73.3	5.08 m
10	51.6	2.53 dd (15.9, 5.4) 2.83 dd (15.9, 7.4)	51.5	2.53 dd (15.9, 5.4) 2.84 dd (15.9, 7.3)	45.5	1.84 m, 1.58 m	42.5	1.97 m 1.68 m
11	72.4	4.34 m	72.3	4.34 m	74.3	4.10 m	72.7	3.97 dq (7.8, 6.0)
12	20.3	1.20 d (6.2)	20.3	1.20 d (6.2 )	20.1	1.21 d (6.0)	20.0	1.19 d (6.0)
Glc-1′	102.4	4.34 d (7.8)	102.3	4.34 d (7.7)	102.3	4.36 d (7.8)	101.8	4.32 d (7.7)
Glc-2′	75.0	3.12 m	75.0	3.12 dd (9.2, 7.8)	75.1	3.15 dd (9.1, 7.8)	75.1	3.14 dd (8.8, 7.7)
Glc-3′	77.8	3.25 m	77.8	3.25 m	77.9	3.28 m	77.8	3.25 m
Glc-4′	71.7	3.26 m	71.7	3.26 m	71.7	3.28 m	71.7	3.30 m
Glc-5′	78.0	3.35 m	78.0	3.36 m	78.0	3.37 m	78.0	3.34 m
Glc-6′	62.9	3.65 dd (11.9, 5.3) 3.84 dd (11.9, 1.9)	62.9	3.64 dd (11.9, 5.3) 3.84 dd (11.9, 1.9)	62.9	3.67 dd (11.8, 5.5) 3.87 dd (11.8, 1.7)	62.9	3.68 dd (11.8, 5.4) 3.79 dd (11.8, 2.3)
C=O							173.0	
CH_3_							21.3	2.01 s

**Table 3 molecules-28-03928-t003:** Results of the antioxidant activity assays of compounds **1**–**7** from the title plant (mean ± SD, *n* = 3).

Compound	IC_50_ (μM)	
	DPPH	ABTS
**1**	48.66 ± 0.94	13.99 ± 2.53 ^b^
**2**	53.85 ± 1.17	13.11 ± 0.94 ^b^
**3**	>100	28.85 ± 0.18
**4**	43.95 ± 1.91	28.44 ± 3.86 ^c^
**5**	>100	33.04 ± 1.43
**6**	>100	38.10 ± 3.94
**7**	>100	27.03 ± 0.55 ^c^
L-ascorbic acid ^a^	30.41 ± 1.40	23.51 ± 0.44

^a^ Positive control. ^b^ The DPPH and ABTS free radicals scavenging abilities of the compound are stronger than the positive control (*p* < 0.05). ^c^ The DPPH and ABTS free radical scavenging abilities of the compounds are equivalent to the positive control (*p* > 0.05).

## Data Availability

The data presented in this study are available in the Appendix A.

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
