# Peer review of "Seven New Phenylhexanoids with Antioxidant Activity from Saxifraga umbellulata var. pectinata"

_molecules, 2023, doi:10.3390/molecules28093928_

Round 1

Reviewer 1 Report

The article presents as a novelty the identification of seven components of the plant Saxifraga umbellata. The authors justify the importance of this work because this plant has been used in traditional Tibetan medicine for the treatment of hepatitis, cholecystitis and digestive diseases, and they want to know the components of the plant that attribute the benefit. The study focuses only on analyzing its antioxidant activity by DPPH and ABTS methods, and evidences that some of the identified components have a higher antioxidant activity than their positive control.  However, the manuscript requires improvements in:

1. introduction, expanding information on the effect of consumption of the plant and how it is prepared for the treatment of these diseases. Also relate the benefit of the antioxidant mechanism of this plant or its components in the context of these diseases, only mention the evidence of an in vitro study with HepG2 cancer cells or antibacterial activity, is that there is no evidence with models of digestive diseases, cholecystitis or hepatitis?

2. In the results and discussion section, it lacks an analysis or discussion of the mechanisms of antioxidant activity that are analyzed with these ABTS and DPPH methods, and why these mechanisms could be involved in counteracting oxidative stress processes and inflammation in these diseases.

3. The ABTS and DPPH assays are chemical methods but they do not demonstrate antioxidant activity in a biological model, therefore, the fact that they are antioxidant with these methods does not mean that they will behave in a biological context, and their relationship with a condition of inflammation associated with oxidative stress as occurs in hepatitis, cholecystitis and some digestive diseases.

4. The article may have interest in the academic community of chemical sciences, but in health sciences, biomedical or biological sciences it has no appeal and is an article similar to many whose scope is only the identification of new molecules, with a potential application in health, but the article lacks that call or inspiration to other researchers to support this type of studies.

5. Finally, Figure 3 can be removed from the article, as it does not contribute anything different from Table 3. That is, either the results of Table 3 are presented in two graphs, both for ABTS and DPPH or the table is presented. I suggest it is more complete and clearer the data in the table, the graph is didactic but it is not possible to know the exact values with which it was constructed.

Author Response

Thank you for your review of our paper. We have answered each of your points below.

Point 1: Introduction, expanding information on the effect of consumption of the plant and how it is prepared for the treatment of these diseases. Also relate the benefit of the antioxidant mechanism of this plant or its components in the context of these diseases, only mention the evidence of an in vitro study with HepG2 cancer cells or antibacterial activity, is that there is no evidence with models of digestive diseases, cholecystitis or hepatitis?

Response 1: Thank you for your suggestion. We have added as much information about the plant as possible in the first paragraph of the introduction and it marked in red. Saxifraga umbellulata var. pectinata is a Tibetan medicine, and research on it has mainly focused on plant identification and the development of quality standards. Due to its remarkable effectiveness in the treatment of liver-related diseases in Tibetan medicine, it has attracted the interest of scholars in recent years to study it systematically. However, studies have mainly focused on its chemical composition, and the exploration of its biological activity is still very limited, so no models of digestive diseases, cholecystitis or hepatitis have been found yet. It is believed that the compounds with novel skeleton mentioned in this paper will arouse the interest of scholars in exploring their biological activities, and we will continue to screen for more biological activities.

Point 2: In the results and discussion section, it lacks an analysis or discussion of the mechanisms of antioxidant activity that are analyzed with these ABTS and DPPH methods, and why these mechanisms could be involved in counteracting oxidative stress processes and inflammation in these diseases.

Response 2: We have only evaluated the in vitro free radical scavenging ability of these compounds using chemical methods, and the mechanism of how these compounds are involved in antioxidant activity in vivo is another process that we have not yet investigated. The rest of the information we have revised and added in the manuscript. Modified section in Line 305-321, marked in red.

Point 3: The ABTS and DPPH assays are chemical methods but they do not demonstrate antioxidant activity in a biological model, therefore, the fact that they are antioxidant with these methods does not mean that they will behave in a biological context, and their relationship with a condition of inflammation associated with oxidative stress as occurs in hepatitis, cholecystitis and some digestive diseases.

Response 3: As you mentioned we have only demonstrated the free radical scavenging ability of the compounds chemically and not in biological models. In this article, we only discuss the in vitro antioxidant activity of these new compounds, and the in vivo antioxidant activity and anti-inflammatory activity of these new compounds are beyond the scope of our discussion. This is only a speculation based on the relationship between oxidative stress and inflammation.. This speculation will hopefully generate more interest for related studies and we will also conduct more in-depth studies.

Point 4: The article may have interest in the academic community of chemical sciences, but in health sciences, biomedical or biological sciences it has no appeal and is an article similar to many whose scope is only the identification of new molecules, with a potential application in health, but the article lacks that call or inspiration to other researchers to support this type of studies.

Response 4: As you said, these seven compounds with novel skeleton do attract great interest in the chemical community, but they will certainly attract scientists in the future because of their special skeleton and potential biological activity. As we know, the development of natural drugs starts with the discovery of compounds and the determination of their structures, so our discovery of these 7 new skeletal compounds is very important. Even though the activity studies of these compounds are not expanded now, it does not mean that no one will study them later. For example, about half a century passed from the discovery of paclitaxel from plants in the 1960s until Abraxane, a drug based on it, was approved by the US FDA for breast cancer treatment in 2005. During that time, it attracted the interest of many scientists to study it.

Point 5: Finally, Figure 3 can be removed from the article, as it does not contribute anything different from Table 3. That is, either the results of Table 3 are presented in two graphs, both for ABTS and DPPH or the table is presented. I suggest it is more complete and clearer the data in the table, the graph is didactic but it is not possible to know the exact values with which it was constructed.

Response 5: Thanks for your suggestion. Figure 3 has been deleted.

Reviewer 2 Report

The manuscript titled "Seven New Phenylhexanoids with Antioxidant Activity from Saxifraga umbellulata var. pectinata" by J. Huang et al. describes the isolation of seven new chemical compounds from plant material. The structures of the isolated compounds were correctly determined. A copy of one- and two-dimensional NMR spectra, FTIR and HR-MS spectra is shown. The antioxidant activity of the isolated compounds was also determined. The overall impression is that the manuscript was carefully prepared and thought out. 

I have the following comments on the manuscript. Please refer to them.

1) Problems with highlighting references. l. 21: Square brackets are missing. l. 74: A system message is displayed. l. 157: The square bracket is missing.

2) Each manuscript object like table, figure, scheme should be mentioned in the text. Figure 1 is left without reference in the text. 

3) l. 67: The text directs to Figure 2 but this drawing does not appear until two pages further down. Manuscript objects (tables, figures, etc.) should be placed as close as possible to mentioning them in the text. 

4) The title of subsection 2.1. "compounds 1-10" is incorrectly stated, in fact seven compounds are described.

5) The same scheme of the text was used to describe how the structure of each compound is described. This way of writing is quite monotonous and tedious to read.

6) The symbols indicating the isomers of disubstituted benzene (o, m and p) should be written in italics. l. 113 and in many other places.

7) l. 114: "trans-double bond." This is a rather unfortunate term. Maybe use: double bond with trans geometry?

8) l. 155 and many other places. Symbols denoting configuration series (D and L) should be written in a much smaller font. 

9) The term "b-D-glucose" is unfortunate, although very common in the literature. The designation of the anomeric hydroxyl group configuration only makes sense when it exists, that is, when the sugar is in ring form. The word "D-glucose" is too general in meaning, and can also refer to the chain form of the sugar. In my opinion, it is better to indicate the type of anomer (alpha or beta) then when you specify the sugar ring. In this case, I would suggest using the term "b-D-glucopyranose".

10) l. 165 "b-D-glucose group". The sugar fragment should not be called a group, rather a moiety. 

11) l. 237 and 266 Please explain for what specific reasons the absolute configuration could not be determined. 

12) Figure 3 shows graphically some of the results placed in Table 3. There should be no duplication in showing the results. Please delete Figure 3.

13) Table 3. Results and their uncertainties have the same decimal expansion and this is in accordance with the rules. In contrast, it is inconsistent with the guidelines to give more than two significant digits for uncertainties. Most uncertainties given have three significant digits. 
